# Sample Preparation Strategies for Antibody-Free Quantitative Analysis of High Mobility Group Box 1 Protein

**DOI:** 10.3390/ph14060537

**Published:** 2021-06-03

**Authors:** Ingeborg Kvivik, Grete Jonsson, Roald Omdal, Cato Brede

**Affiliations:** 1Research Department, Stavanger University Hospital, 4011 Stavanger, Norway; ingeborg.kvivik@sus.no; 2Department of Medical Biochemistry, Stavanger University Hospital, 4011 Stavanger, Norway; grete.jonsson@sus.no; 3Department of Clinical Science, Faculty of Medicine, University of Bergen, 5020 Bergen, Norway; roald.omdal@sus.no; 4Clinical Immunology Unit, Department of Internal Medicine, Stavanger University Hospital, 4011 Stavanger, Norway; 5Department of Chemistry, Bioscience and Environmental Engineering, University of Stavanger, 4021 Stavanger, Norway

**Keywords:** antibody-free, blood, chromatography, HMGB1, mass spectrometry, sample preparation

## Abstract

Sickness behavior and fatigue are induced by cerebral mechanisms involving inflammatory cytokines. High mobility group box 1 (HMGB1) is an alarmin, and a potential key player in this process. Reliable quantification methods for total HMGB1 and its redox variants must be established in order to clearly understand how it functions. Current methods pose significant challenges due to interference from other plasma proteins and autoantibodies. We aimed to develop an antibody-free sample preparation method followed by liquid chromatography coupled with tandem mass spectrometry (LC-MS/MS) to measure HMGB1 in human plasma. Different methods were applied for the removal of interfering proteins and the enrichment of HMGB1 from spiked human plasma samples. A comparison of methods showed an overall low extraction recovery (<40%), probably due to the stickiness of HMGB1. Reversed-phase liquid chromatography separation of intact proteins in diluted plasma yielded the most promising results. The method produced an even higher degree of HMGB1 purification than that observed with immunoaffinity extraction. Detection sensitivity needs to be further improved for the measurement of HMGB1 in patient samples. Nevertheless, it has been demonstrated that a versatile and fully antibody-free sample preparation method is possible, which could be of great use in further investigations.

## 1. Introduction

High mobility group box 1 (HMGB1) protein is a highly conserved protein that participates in the organization of DNA and gene transcription in the nucleus [1]. In addition, it is released into the cytoplasm during cellular stress, and can also be secreted from immune cells and can strongly promote inflammation [2]. In the latter, HMGB1 functions as a danger-associated molecular pattern (DAMP) protein, an alarmin or a sensor of cellular harm and initiates a strong activation of innate immune responses through interactions with receptors for advanced glycation end-products (RAGE), as well as Toll-like receptors (TLR)-2 and -4 [3]. The protein also has the ability to bind to other proinflammatory molecules when they coexist with HMGB1, thus enhancing the activation of immune reactions and inflammation [4].

HMGB1 is a relatively small protein, having 215 amino acid residues, with a molecular weight of 25 kDa. Depending on the clinical conditions, various redox isoforms of the molecule appear and possess different signaling properties. During inflammation, disulfide (ds) and fully reduced (fr) HMGB1 strongly stimulate TLR4 and RAGE, respectively, leading to the production and release of several proinflammatory cytokines [5].

Proinflammatory cytokines (especially interleukin (IL)-1) are inducers of sickness behavior, observed in animals and humans during infection and states of chronic inflammation or cellular stress. Fatigue is a major component of sickness behavior; it is well known that IL-1 is a key player in the induction of sickness behavior through interactions with specific IL-1 receptors on cerebral neurons [6,7]. Based on our longstanding study of chronic fatigue [8,9,10,11], we postulated that HMGB1 could be an important signaling molecule in the context of sickness behavior. Such a behavior is induced via the activation of cerebral mechanisms; hence, cerebrospinal fluid (CSF) is the preferred sample material for quantitative analyses. However, spinal puncture is more complicated than venipuncture; hence, blood is a more practical sample material for such studies. In this report, we describe the experimental difficulties encountered in performing a valid antibody-free assay for measuring HMGB1 in plasma.

HMGB1 can be measured in blood plasma and serum via western blotting (WB) or enzyme-linked immunosorbent assay (ELISA). Even though immunoassays for HMGB1, such as ELISA, are highly sensitive and have high throughput, they are prone to errors caused by the presence of HMGB1 autoantibodies, cross-reactivity interference, or competitive binding of other proteins present in blood [12,13]. Recently, preanalytical conditions were found to have a significant effect on HMGB1 measurements obtained through ELISA [14]. Furthermore, knowledge about the redox variants of HMGB1 with different physiological functions has created a requirement for distinguishing between reduced and oxidized HMGB1. These issues suggest that there is an unmet need for accurate and alternative methods for measuring HMGB1.

Liquid chromatography coupled with tandem mass spectrometry (LC-MS/MS) provides an alternative to immunoassays, and is frequently implemented in clinical laboratories [15]. In the assessment of protein concentrations by LC-MS/MS, it is common to apply enzymatic cleavage of proteins and measure the concentrations of the corresponding proteotypic peptides. LC-MS/MS provides multiple dimensions for the selective detection of HMGB1, including retention time, as well as molecular ion and fragment ion masses (*m/z*) as identification parameters. Recent advancements in LC-MS/MS methods for HMGB1 have implemented immunoaffinity extraction (IAE) to isolate either intact HMGB1 [16,17,18] or peptides derived from HMGB1 after enzymatic cleavage [19]. The former approach suffers from interference by various binding proteins, such as autoantibodies, whereas the latter approach only catches part of the protein and does not allow for the full study of post-translational modifications (PTMs) [20]. However, we still do not know the extent to which HMGB1 is acetylated, phosphorylated, or oxidized in vivo. In addition, we do not know how these modifications may affect the efficiency of IAE. This issue was elegantly addressed by Weng et al. [17], where both the analyte and internal standard were subjected to full acetylation before IAE. However, no antibody-free LC-MS/MS method has been reported for the measurement of HMGB1 in human blood samples. It is evident that HMGB1 was easily lost in various steps during sample cleanup. Hence, our approach involves a substantial number of trials and errors. The present paper reports different antibody-free sample preparation strategies for HMGB1, with the aim of removing interfering molecules, detecting original redox modifications, and being applicable for total HMGB1 quantification in blood.

## 2. Results and Discussion

### 2.1. Detection of HMGB1 Peptides

Proteotypic HMGB1 peptides from samples of pure recombinant HMGB1 were studied using LC-MS/MS with high-resolution mass spectrometry (HRMS). Reconstructed extracted ion chromatograms (EIC) for the peptides were created, with a 5 mDa tolerance at the theoretical *m*/*z* of the peptide molecular ions [M + 2H]^2+^, as shown in Figure 1. The peptides were identified from their fragmentation mass spectra via sequence modeling using the Global Proteome Machine (GPM) software and the human Uniprot dataset downloaded in FASTA format. A typical MS/MS fragmentation mass spectrum is shown in Figure 2. The most prominent peptides identified from recombinant HMGB1 were IKGEHPGLSIGDVAK (IKG), with [M + 2H]^2+^ at *m*/*z* 760.925, KHPDASVNFSEFSK, with [M + 2H]^2+^ at *m*/*z* 796.889, and the MSSYAFFVQTCR peptide (MSS) alkylated with N-ethylmaleimide (NEM), with [M + 2H]^2+^ at *m*/*z* 782.850. Methionine (amino acid 13) in MSS peptides was often found to be oxidized, resulting in a mass shift of 15.995 Da; hence, both oxidized ([M + 2H]^2+^ at *m*/*z* 790.846, Appendix A) and non-oxidized MSS were included in further analyses. MSS contains a cysteine residue at amino acid position 23 (C23), which forms a disulfide bridge in dsHMGB1 with the cysteine at position 45 (C45). This peptide is important for determining the original oxidation status of HMGB1 via differential alkylation [21] of the cysteines C23 and C45 with NEM and N-methylmaleimide (NMM), resulting in mass shifts of 125.048 Da and 111.032 Da, respectively. The most frequent HMGB1 peptides identified after trypsin digestion of recombinant HMGB1 are shown in Appendix A. The instrumental limit of detection (LOD) for HMGB1 was investigated by integrating the peak area for IKG in the EIC using pure HMGB1 tryptic digests, and was found to be 5 ng/mL.

### 2.2. Sample Preparation

#### 2.2.1. Challenges and Alternative Strategies

Detecting trace concentrations of HMGB1 in human plasma is a typical “needle in a haystack” challenge. It requires a fine balance between removing the majority of the interfering proteins, while preventing the loss of the analyte. Complex matrices, such as serum and plasma, need to be cleaned before LC-MS/MS analyses. Unfortunately, gel electrophoresis is not suitable for crude serum samples [22]. Furthermore, the concentration of HMGB1 in blood is very low compared to twenty-two of the high-abundance proteins (HAP) (including albumin and gamma globulins) which make up 99% of the plasma proteins [23], and may mask the signals from HMGB1 in the LC-MS/MS analysis. The ideal sample preparation should be able to enrich HMGB1 in the final sample extract, reduce the amount of interference to a minimum, and preserve the original PTMs. The removal of HAPs also risks the depletion of HMGB1 because of its many binding partners.

Data-dependent acquisition (DDA) allowed for the simultaneous study of both HMGB1 recovery and the presence of HAP interference, using the number of matching peptides (spectral counting) by the GPM search to semiquantitatively rank identified proteins based on their abundance. We used this ranking to indicate the extent of sample cleanup. An increased number of identified peptides from other proteins that remained in the samples forced HMGB1 to a lower ranking position in the list of identified proteins. In contrast, HMGB1 was ranked higher, with more unique HMGB1 peptides identified. Therefore, a high ranking of HMGB1 among all proteins served as a strong indicator of successful sample preparation. Recovery experiments were carried out by spiking HMGB1 into human plasma and subsequently measuring the integrated peak area response for either MSS or IKG in their respective EICs relative to the response in solutions of pure recombinant HMGB1.

In general, three different sample preparation strategies were compared, as illustrated in Figure 3: (A) targeted extraction of HMGB1 from the sample, (B) precipitation of the majority of HAPs while HMGB1 is kept in solution, and (C) processing the whole sample followed by intact peptide or protein fractionation by high performance liquid chromatography (HPLC).

#### 2.2.2. Targeted Extraction (A)

IAE is a relatively common technique for capturing low-abundance proteins (such as HMGB1) from biological samples, including human plasma [24,25,26]. Similar to a recently reported method [17], we coupled anti-HMGB1 antibodies to magnetic beads and applied them for IAE of HMGB1 from plasma. The aim was to bind HMGB1 to the beads and allow HAP interference to be washed away prior to the elution of HMGB1 from the beads. On-bead trypsin digestion was also attempted to reduce the potential loss of HMGB1 caused by irreversible adsorption [27]. Several commercial antibodies were tested (data not shown), the most promising results being obtained with anti-HMGB1 antibodies from Abcam (# Ab18256). However, the recovery was found to be as low as 7%, due either to the low antibody binding of HMGB1 or the fact that autoantibodies or other HMGB1-binding plasma proteins prevented a more quantitative binding of HMGB1 to the immobilized antibody. The results of protein identification revealed high amounts of other plasma proteins captured and released with HMGB1 (Table 1). We followed up on this experiment via the pretreatment of plasma samples with immobilized protein-G (IgG depletion), but unfortunately this produced even further reduction in recovery. Initially, the idea of using IAE likely offered a basis for comparing other sample preparation techniques. However, it was concluded that IAE with anti-HMGB1 antibodies provided neither clean sample extracts nor a sufficiently high recovery of HMGB1.

A second approach for the targeted extraction of HMGB1 was to capture haptoglobin. It is known that haptoglobin forms a complex with HMGB1 [31] (particularly dsHMGB1) with potential anti-inflammatory properties. Therefore, we investigated whether HMGB1 could be captured by targeting haptoglobin. Magnetic beads coupled to anti-haptoglobin antibodies were applied for IAE, which managed to capture haptoglobin and HMGB1 from plasma spiked with dsHMGB1. These experiments resulted in a slightly higher (16%) recovery of HMGB1 compared to anti-HMGB1 antibodies. In this experiment, only two unique HMGB1 peptides were identified (Table 1). Interestingly, apolipoproteins were identified in addition to haptoglobin and HMGB1 in these extracts. This corresponded well with the fact that haptoglobin is known to bind apolipoproteins, including apolipoprotein A-I [32]. The low recovery can be explained by a weaker-than-expected binding of HMGB1 to haptoglobin, or by competition with apolipoproteins. It is also possible that the IAE procedure was not fully optimized for capturing haptoglobin, or was followed by inefficient trypsin digestion. Thus, we abandoned this technique. However, the observation of somewhat clean sample extracts was encouraging for future work in this direction.

#### 2.2.3. Precipitation of HAPs (B)

Because of HMGB1 being a rather small protein, our initial idea for HAP removal was with acetonitrile (ACN) precipitation, as reported for human serum [33] and cerebrospinal fluid [34]. We soon abandoned ACN precipitation because it resulted in nondetectable levels of HMGB1, likely due to coprecipitation with other proteins or poor solubility at the higher concentration levels of ACN that were required for HAP removal. One of the special features of HMGB1 and other HMG-family proteins is that they are soluble in diluted perchloric acid (PCA) [35]. This property has been successfully utilized in several previously reported methods [36,37]. We found this approach to potentially offer some advantages for the isolation of HMGB1 from plasma proteins. Precipitation was induced by adding 13.7% PCA to plasma samples to a final concentration of 5% PCA, leaving HMGB1 soluble in the supernatant. However, the acidic supernatant needs to be neutralized or removed before enzymatic cleavage. The buffer was changed using 5 kDa molecular weight cut-off (MWCO) spin filters. HMGB1 and proteins with MW >5 kDa were retained and could be repeatedly rinsed while remaining on the filter. This allowed for filter-aided sample preparation (FASP) for reduction, alkylation, and trypsin digestion [38]. Peptides with MW <5 kDa were collected in the flow-through below the filter. This method produced a recovery of up to 35%, resulting in six unique HMGB1 peptides identified in the spiked plasma (Table 1). However, the MSS-peptide was not identified, likely due to loss through adsorption onto the filter surface or due to suboptimal cleavage. This approach was not pursued further, as MSS was regarded as important for identifying redox variants. With this method, HMGB1 was ranked as number 28 among the 61 proteins identified, which strongly indicated the need for more efficient HAP removal.

The historical success of using dilute PCA for maintaining HMGB1 in solution is probably due to the higher initial concentrations of the protein in previously reported experiments. This method has traditionally been used for processing tissue homogenates containing high amounts of HMGB1 from the nucleus [36,39]. HMGB1 binds to many other proteins, and a fraction of HMGB1 likely coprecipitated with HAPs in the initial step. At higher concentrations of HMGB1, such losses may be less evident than in our experiments, which have a very low concentration of HMGB1 spiked into the plasma. There is also a possibility that the low pH of PCA affects the oxidation status of the protein, resulting in compromised information about the original redox status of HMGB1 [40]. These effects can be prevented via reduction and alkylation before PCA precipitation. In future works, it would be worth pursuing better solubility for the proteotypic peptides, as well as investigating spin filters of different materials and cut-off ranges.

#### 2.2.4. Processing Whole Samples (C)

The concept of whole sample preparation would theoretically avoid all issues associated with complex binding and interfering antibodies. The enzymatic cleavage of a sample before LC-MS/MS analysis would enable the measurement of total HMGB1 concentrations, irrespective of the originally bound or free HMGB1. However, the enzymatic cleavage of neat plasma results in a vast number of peptides from HAPs, which makes it necessary to fractionate the sample extracts before LC-MS/MS analysis. We experimented with the nonenzymatic cleavage of plasma proteins using acetic acid, similar to formic acid cleavage as previously reported [41], and later modified to use acetic acid to prevent formylation [42]. The reaction was performed at a high temperature, which enabled the thermal denaturation of the protein in addition to cleavage at the C-terminal of aspartic acid. HMGB1 has 20 such residues, most of which are located at the C-terminal, resulting in a few, rather large peptides that were directly separated by HPLC (Appendix A). Several fractions obtained from HPLC were collected, digested a second time with trypsin, and analyzed via HRMS to locate peptides specifically from HMGB1. Only a limited portion of HMGB1 was covered by sequence identification of these large peptides; fortunately, a few also covered complete MSS and partial IKG peptide sequences (Appendix A). These peptides were found in fractions between 19–25 min.

Fine tuning of fraction sampling resulted in four 2-min fractions between 17.8–25.8 min. Different HMGB1 proteotypic peptides were found in different fractions (Appendix A). We decided to focus on the 19.8–21.8 min fraction, where the MSS peptide was found. The method could recover ~30% of neat HMGB1, but when applied to spiked plasma samples, the recovery decreased to 12%. HMGB1 was ranked 51 among the 68 proteins identified in this fraction (Table 1). The acid digestion of plasma was likely suboptimal, as many of the peptides found after both acid and trypsin digestion indicated improper cleavage (Appendix A). Further work in this direction should attempt to use more diluted plasma samples to ensure a constant low pH during digestion.

The second whole sample processing approach involved the fractionation of intact proteins via HPLC. High mobility group proteins have previously been separated by reversed-phase LC using a C18 stationary phase [43,44]. Similarly, we applied a wide pore C18 column to separate HMGB1 from other plasma proteins, using an ACN gradient and ion-pairing with trifluoroacetic acid (TFA). We found it feasible to inject pure recombinant HMGB1 dissolved in 10% formic acid (FA); with 0.1% TFA in the mobile phase, the protein was eluted by 30% ACN between 9.5–11 min. We then applied plasma spiked with recombinant HMGB1 and diluted with water, along with 25% FA, which resulted in the majority of other proteins in the sample being eluted at higher ACN concentrations. Hence, we were able to achieve direct HPLC fractionation of HMGB1 from most other plasma proteins. Several protein fractions were collected from the column before a massive protein peak at 12 min, eluting with >40% ACN in the mobile phase (Figure 4). The fractions were then dried via evaporation in a vacuum centrifuge, trypsin-digested, and, finally, analyzed using HRMS. The proteins in each fraction were identified. As many as 14 unique HMGB1 peptides were identified in the fractions collected between 9.5–11 min. HMGB1 was ranked as the third most abundant protein among 19 other plasma proteins (Table 1), which indicated a rather high degree of purification. The recovery was 33% for this sample preparation method. Small traces of HMGB1 could also be found in subsequent fractions, thus, indicating a certain tailing of proteins on the column. In future work, it might be worth pursuing the optimization of the mobile phase, or applying a more deactivated stationary phase to further increase recovery. One advantage of preparing the intact protein by HPLC is that it can be combined with diverse preanalytical treatments and digestion methods. For example, reduction and alkylation to identify the original redox status of HMGB1 need to be performed before the sample is exposed to low pH with the addition of FA. Alkylation with NEM or NMM did not alter the retention time of the protein. Hence, this method permits the study of HMGB1 redox status.

### 2.3. Future Perspectives

#### 2.3.1. Method Improvement

HPLC fractionation of intact HMGB1 from other plasma proteins was by far the most promising of all sample preparation methods that we compared, despite a somewhat low recovery. With an extraction recovery of only approximately 30%, it is extremely important to include an internal standard (ISTD) at an early stage of sample preparation. To some extent, the ISTD can compensate for losses during sample preparation. Fortunately, isotopically labeled intact HMGB1 is now commercially available (HMGBiotech, Milan, Italy), and provides a unique opportunity for implementing high-quality internal standard quantification in future LC-MS/MS methods [45]. Adding a known amount of fully ^15^N-labeled internal standard immediately before sample preparation should make it possible to quantify HMGB1 using peptides derived from it via enzymatic cleavage, and calculate the ratios between peptides from HMGB1 and ^15^N HMGB1. The use of an internal standard would also eliminate sample variations in enzymatic cleavage.

At the same time, the LOD for instrumental analysis must be improved. This may be achieved by using more sensitive MS instrumentation (such as multiple reaction monitoring (MRM) with a sensitive triple-quadrupole MS) in combination with a downscaled LC separation. So far, trials with implementation of the internal standard combined with a more sensitive MRM method has not been used for measuring HMGB1 in any patient sample, although we observed an improved instrumental LOD of 1.5 ng/mL compared to 5 ng/mL with HRMS. Two successful reports on the measurement of HMGB1 in human samples applied nano-LC-MS/MS to achieve adequate detection. They found plasma concentrations in healthy subjects to be less than 1 ng/mL [17,19]. Hence, more work is needed to improve the LOD, perhaps facilitated by new and improved MS detection. In the future, we hope to further reduce the LOD by combining a downscaled LC separation (micro-or nano-LC) with MRM or a more sensitive HRMS detection (Orbitrap).

#### 2.3.2. Choice of Sample Preparation

Several PTMs of HMGB1 (such as phosphorylation, glycosylation, methylation, and ADP-ribosylation) have been described (reviewed in [46]), including acetylation of lysine residues as well as the oxidation of cysteine residues. In fact, it is this redox status that affects its immunological activity, and information about acetylation identifies the origin of the protein, and whether it is secreted actively or passively. Hence, the choice of sample preparation method needs to be considered carefully to enable the preservation or tagging of the PTMs.

The binding partners of HMGB1 are of interest as well. We can assume that many of the HMGB1 molecules in blood are bound in some way, possibly for the purpose of transport, signaling together with a binding partner, or downregulation via antibody-binding [4,31,47,48]. The addition of FA to the samples before HPLC separation makes HMGB1 more soluble, and separable from HAPs with the C18 column. However, information on whether HMGB1 in vivo is bound or unbound is lost in the process. These parameters need to be considered when investigating the role and function of this protein in vivo. The discussion conclusively suggests that the chosen sample preparation strategy will indeed affect the type of information obtained.

## 3. Materials and Methods

### 3.1. Materials

#### 3.1.1. Chemicals

LC-MS grade ACN was purchased from VWR Chemicals BDH (Radnor, Pennsylvania, USA). HPLC grade >98% FA and >99% TFA, 37% hydrochloric acid (HCl), 70–72% PCA, Trizma base, DL-dithiothreitol (DTT), NEM, NMM, 25% ammonium hydroxide, Tween, and ammonium bicarbonate (AMBIC) were purchased from Merck (Darmstadt, Germany). Ethanol (99.9%, EtOH) was purchased from Antibac (Asker, Norway). Glycine >99%, iodoacetamide (IAA), and CaCl_2_ were purchased from Sigma-Aldrich (St. Louis, MO, USA). Phosphate-buffered saline (PBS) was obtained from Thermo Fisher Scientific (Waltham, MA, USA). Crystalline porcine trypsin (4500 K) was obtained from Novozymes A/S (Bagsværd, Denmark). Recombinant HMGB1 for biochemistry and dsHMGB1 were obtained from HMGBiotech S.r.l. (Milano, Italy). Anti-HMGB1 antibodies (Ab18256, polyclonal, rabbit) were purchased from Abcam (Cambridge, UK). Antihuman haptoglobin antibodies from immunoturbidimetry clinical chemistry reagents (Roche Diagnostics, Rotkreuz, Switzerland) were purified to remove the tris-buffer by three rounds of dialysis with 0.1% sodium chloride, using cellulose membrane dialysis tubing of approximately 14k MWCO (Merck, Darmstadt, Germany).

#### 3.1.2. Plasma Samples

Plasma from healthy volunteers was anonymized and used for the development of the method. Blood was drawn on EDTA sample tubes and centrifuged for 15 min at 2800× *g* at 4 °C to separate the plasma from blood cells. If the plasma was not used immediately, it was aliquoted and frozen at −70 °C.

### 3.2. Methods

#### 3.2.1. Separation of Proteins Using HPLC

Plasma samples were diluted with FA and water (1 + 1 + 2), and 100 µL (full loop) was injected using an Alliance 2795 HPLC system (Waters, Milford, MA, USA) onto a protein separation column (ACE 5 C18-300, 250 mm × 4.6 mm) maintained at 40 °C in a column oven. A gradient was run with a mobile phase flow rate of 0.6 mL/min, and solvent A (0.1% TFA) was mixed with solvent B (ACN) as follows: 10% solvent B for 2 min, 10–30% B from 2–5 min, 30–70% B from 5–22 min, 70–90% B from 22–23 min, and 90% B from 23–30 min (Figure 4). The UV detector (Waters 996 Photodiode Array Detector, Waters, Milford, MA, USA) was set at 280 nm. Fractions were collected between 9.5–11 min, in conical 1.5 mL polypropylene tubes, using a fraction collector (Advantec SF2100W, Advantec MFS, Dublin, CA, USA). The HPLC system was controlled using MassLynx v. 4.1 (Waters) software.

The fractions were dried using a miVac concentrator with Speed Trap (Genevac Ltd., Ipswich, UK) at 80 °C for 2 h. Then, three fractions from the same sample were pooled and evaporated to complete dryness.

#### 3.2.2. Reduction, Alkylation, and Trypsin Digestion

Dried protein extracts were resuspended in 20 µL of 100 mM Tris, adjusted to pH 7.7 with FA (Tris-FA). Proteins eluted from IAE or concentrated on a MWCO filter were adjusted to a volume of 20 µL. Reduction was performed by adding 10 µL of 30 mM DTT to the sample and incubating for 10 min at RT. Next, 10 µL of 30 mM NEM in EtOH or 30 mM NMM in EtOH was added to alkylate all cysteine residues for 10 min at RT. Then, 0.1 µg trypsin (or the amount/concentration that gave 1:20 trypsin:protein) with 20 mM CaCl_2_ was added, and the volume was adjusted with Tris-FA to a total of 50 µL or 100 µL, before digestion for 17 h at 37 °C. Digestion was stopped by adding 1 µL TFA.

#### 3.2.3. HRMS

Unique peptides from recombinant human HMGB1 were produced via enzymatic cleavage using trypsin. Peptides were identified by data-dependent acquisition (DDA) after liquid chromatography coupled with tandem mass spectrometry (LC-MS/MS), using an I-Class UPLC (Waters, Milford, MA, USA) coupled with a timsTOF hybrid quadrupole/time-of-flight (QTOF) high-resolution mass spectrometer (Bruker, Billerica, MA, USA). Ionization was achieved via positive electrospray (ESI+) with a capillary voltage of 4.5 kV and a drying gas flow rate of 9 L/min at 200 °C. The molecular ion mass range was 50–1600 m/z, and the mass range for fragment ions in MS/MS mode was 300–1500 m/z.

LC-separation was performed on an Acquity 150 mm × 2 mm BEH C18 column (Waters, Milford, MA, USA) at 45 °C. Ten microliters of the prepared sample was injected. A 30 min gradient with mobile phase A (0.1% ammonium hydroxide) and mobile phase B (ACN) was run at a flow rate of 0.1 mL/min, as follows: 99% A for 1 min, 90% A for 2–20 min, 70% A for 20–21 min, 30% A for 21–25 min, followed by a re-equilibration at 26 min with 0.3 mL/min using 99% A for 4 min.

#### 3.2.4. Protein Identification

Protein identification was achieved by analyzing the data generated in the mascot generic format (MGF) (Matrix Science, Boston, MA, USA) using the 64-bit Windows version of GPM v. 3.0 open-source software [28]. GPM counts peptide fragmentation mass spectra, and identifies peptides in the samples using the X!Tandem search engine [29]. The settings for precursor tolerance were 20 mDa, and a fragment tolerance of 0.4 Da was used. Complete modifications of cysteines were set to 125.0477 Da for alkylation with NEM, and 111.0320 Da for alkylation with NMM. The oxidation and dioxidation of methionine and tryptophan, as well as the acetylation of lysine, were set as potential modifications. The MGF files were uploaded to a data repository [30]. Chromatograms and mass spectra were analyzed using Bruker Compass Data Analysis version 5.0 (Bruker, Billerica, MA, USA).

## 4. Conclusions

HMGB1 is an evolutionarily old protein, with several important roles in genomic organization and function and in innate immunity. At least three isoforms of the protein exist, all of which have different functions. A large body of evidence points to HMGB1 as a modulator of cerebral function and as a partner in inducing sickness behavior and fatigue. Therefore, we attempted to develop and refine a sample preparation method suitable for mass spectrometry analysis, which allows the measurement of total HMGB1, without interference from autoantibodies or other binding molecules. Like others, we have learnt that this can be extremely difficult [49]. The recurring problem with the different methods was low recovery. By removing HAPs, there was always the potential of removing HMGB1 as well, due to the stickiness of the protein. HPLC separation of diluted plasma samples yielded the purest sample extracts compared to the other sample preparation methods. Despite a somewhat suboptimal recovery, this sample preparation method was found to be the most useful. The identification of several unique HMGB1 peptides will enable the measurement of different redox variants as well as other potential PTMs.

We show here that it is possible to measure HMGB1 in plasma using a totally antibody-free sample preparation method. Our work could be a useful advancement for the study of this important alarmin in human samples, such as blood and cerebrospinal fluid.

## Figures and Tables

**Figure 1 pharmaceuticals-14-00537-f001:**
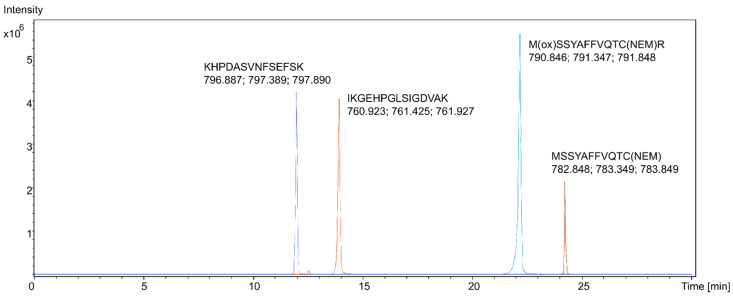
Reconstructed extracted ion chromatograms (EIC) for four specific high mobility group box 1 (HMGB1) peptides. The three masses for each peak represent the theoretical *m*/*z* of the peptide molecular ions [M + 2H]^2+^ and the two most abundant ^13^C isotope signals.

**Figure 2 pharmaceuticals-14-00537-f002:**
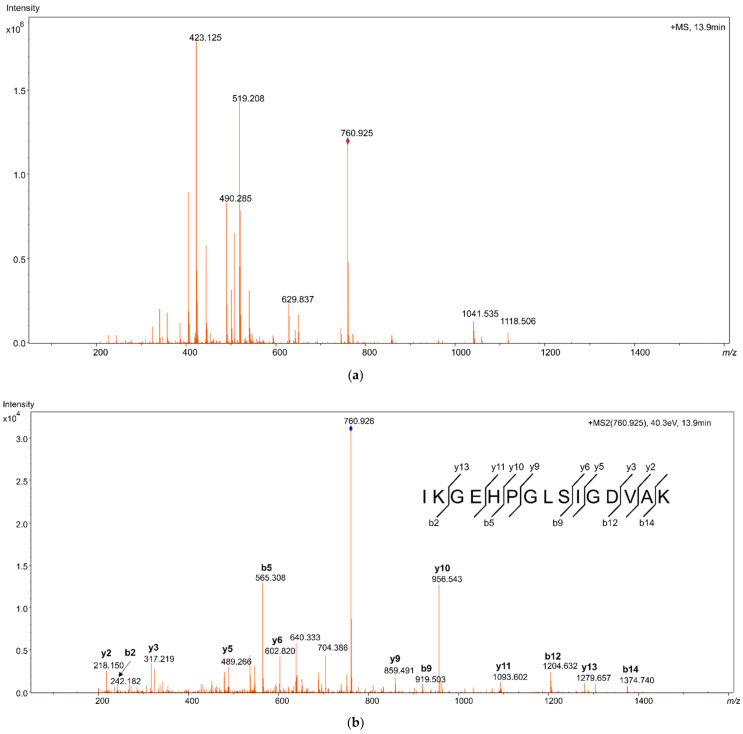
Mass spectrum from high resolution mass spectrometry (HRMS), identifying high mobility group box 1 (HMGB1) peptide IKGEHPGLSIGDVAK (**a**) Molecular ion mass spectrum showing the peptide mass [M + 2H]^2+^ with *m*/*z* 760.9245, at 13.9 min. (**b**) Fragment ion mass spectrum of *m*/*z* 760.9245 showing series of y- and b-ions identifying the amino acid sequence.

**Figure 3 pharmaceuticals-14-00537-f003:**
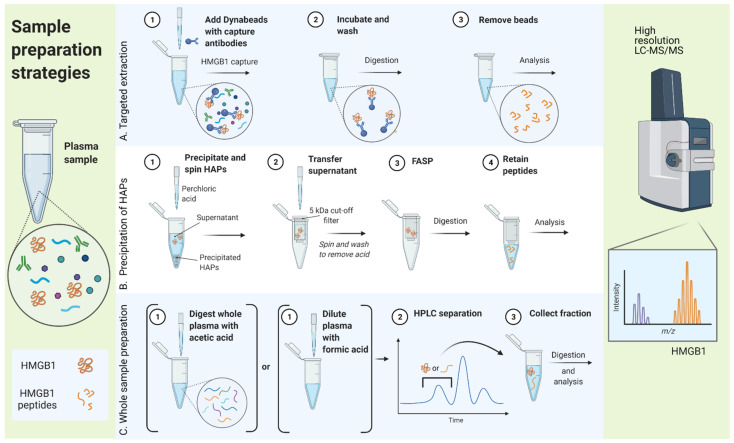
Different sample preparation strategies used to optimize sample clean-up for antibody-free analysis of high mobility group box 1 (HMGB1) in human plasma. FASP, filter aided sample preparation; HAP, high abundance proteins; HPLC, high performance liquid chromatography; LC-MS/MS, liquid chromatography coupled with tandem mass spectrometry. Created with BioRender.

**Figure 4 pharmaceuticals-14-00537-f004:**
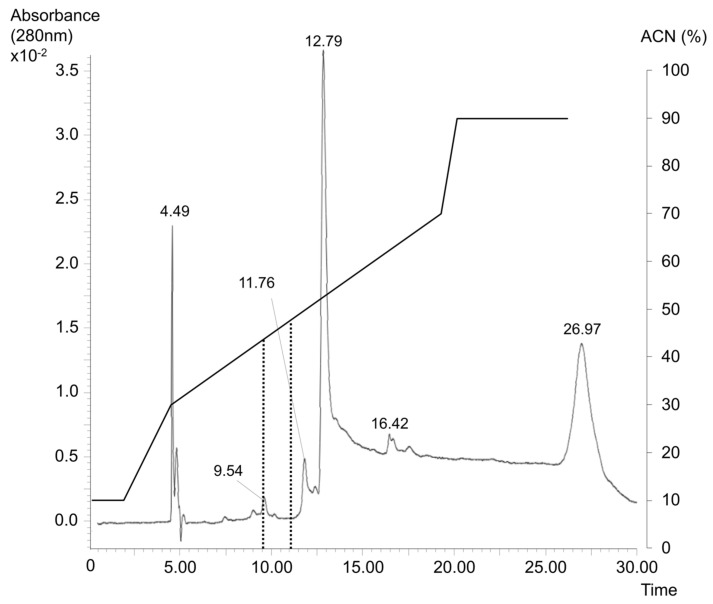
Chromatogram from high performance liquid chromatography (HPLC) of recombinant high mobility group box 1 (HMGB1) spiked in plasma. The gradient of acetonitrile (ACN) is overlaying the chromatogram. The fraction from 9.5–11 min, where HMGB1 is eluted, is marked.

**Table 1 pharmaceuticals-14-00537-t001:** Outcomes of different sample preparation strategies for measuring high mobility group 1 (HMGB1) protein in human plasma. Recovery was calculated by comparing the peak area response from the extracted ion chromatogram for the IKGEHPGLSIGDVAK peptide in processed samples along with the response in samples with pure recombinant HMGB1. The number of peptides and other proteins identified was found by searching the Global Proteome Machine (GPM) database [28,29]. Data files in the mascot generic format (MGF) can be downloaded from a data repository online [30].

Sample Preparation Strategy		Recovery (%)	Results from Search in the GPM Database			File Name (.mgf)
			Number of Specific HMGB1-Peptides	Total Proteins Identified	Rank# for HMGB1	
A	Targeted extraction using anti-HMGB1 Ab	7	None	5	N/A	HMGB1_1
	Targeted extraction using anti-haptoglobin Ab	16	2	5 ^a^	3	HMGB1_2
B	Precipitation of HAPs using PCA	35	6	61	28	HMGB1_3
C	Acid digestion of whole plasma	12 ^b^	2 ^c^	68 ^c^	51 ^c^	HMGB1_4
	Separation of proteins using HPLC	33	14	19	3	HMGB1_5
Pure recombinant HMGB1		100	17	1	1	HMGB1_6

^a^ Contaminants not counted. ^b^ Recovery calculated using peak area response from extracted ion chromatogram for M(ox)SSYAFFVQTCR(NEM) peptide in processed samples compared to pure recombinant HMGB1. ^c^ Fractions between 19.8 min and 21.8 min. Ab, antibody; HAP, high-abundance proteins; HPLC, high performance liquid chromatography; PCA, perchloric acid.

## Data Availability

The data presented in this study are openly available in Mendeley Data at [doi:10.17632/p4wfz6hhf4.1], reference number [30].

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
