# Peer review of "Sample Preparation Strategies for Antibody-Free Quantitative Analysis of High Mobility Group Box 1 Protein"

_pharmaceuticals, 2021, doi:10.3390/ph14060537_

Round 1
Reviewer 1 Report
The manuscript by Ingeborg Kvivik et al., entitled "Strategies for Antibody-Free Quantitative Analysis of High 2 Mobility Group Box 1 Protein", is highly interesting, well written and structured.
The authors also described with rigour the different sample preparation strategies used to optimize sample clean-up for antibody-free analysis of high mobility 160 group box 1 (HMGB1) in human plasma.
I propose the publication of this manuscript in the present form.
Author Response
Thank you for the nice review of our manuscript "Strategies for Antibody-Free Quantitative Analysis of High Mobility Group Box 1 Protein". There were no changes that needed to be done.
Reviewer 2 Report
The submitted manuscript « Strategies for antibody-free quantitative analysis of high mobility group box 1 protein” nicely illustrates the importance of sample preparation prior to MS, in particular for complex matrices such as blood plasma. Three sample preparation alternatives were tested. The approach used is sounded and was carefully carried out and well documented concerning MS analyses. Precipitation using PCA and separation of proteins using HPLC gave the highest recovery, similar for both methods but the number of specific HMGB1-peptides was larger after LC-separation. The authors thus concluded that it is possible to measure HMGB1 in plasma using totally antibody-free sample preparation method.
Questions:
1) Why not add in the title “Sample Preparation strategies…” ?
2) The 2.3 Future perspectives section is very well documented but gives a taste of unfinished work…
Have the authors already tested and have some preliminary data on the potential improvement ideas ? If so, this should be mentioned.
3) Have the authors information on the recovery starting from blood serum instead of plasma ? If so, this should be mentioned.
4) Have the authors tried other precipitation methods prior to LC/MS-MS ? For example :
Ref: Kay, R., Barton, C., Ratcliffe, L., Matharoo-Ball, B., Brown, P., Roberts, J., Teale, P., Creaser, C. (2008) Enrichment of low molecular weight serum proteins using acetonitrile precipitation for mass spectrometry based proteomic analysis, Rapid Commun. Mass Spectrom., 22, 3255–3260.
Link : https://pubmed.ncbi.nlm.nih.gov/18803344/
This method used with pure recombinant HMGB1 will rapidly inform us on the recovery of the target protein in supernatant, and precipitate if any.
5) Line 51 : ‘Based on our longstanding study…’ have the authors any reference on that ?
Author Response
Response to reviewer 2
Thank you for your positive and useful review of our manuscript “Strategies for antibody-free quantitative analysis of high mobility group box 1 protein”. Below follows our answers to your comments/questions:
1) Why not add in the title “Sample Preparation strategies…”?
Response:
Thank you for the suggestion, we think it is a good addition to the title, and have changed it so it now reads “Sample Preparation Strategies for Antibody-Free Quantitative Analysis of High Mobility Group Box 1 Protein”.
2) The 2.3 Future perspectives section is very well documented but gives a taste of unfinished work…
Have the authors already tested and have some preliminary data on the potential improvement ideas? If so, this should be mentioned.
Response:
This is a very relevant question. The uses of isotopic labelled internal standard and MRM detection are mentioned in the manuscript. Other improvement ideas have not been tested. We have changed the sentences in lines 328-330, so that it now reads: “Hence, more work is needed to improve the LOD, perhaps facilitated by new and improved MS detection. In the future, we hope to further reduce the LOD by combining a downscaled LC separation (micro- or nano-LC) with MRM or a more sensitive HRMS detection (Orbitrap).”
3) Have the authors information on the recovery starting from blood serum instead of plasma? If so, this should be mentioned.
Response:
Thanks for noticing this. The sample matrix and the preanalytical conditions are important to consider during method development. Initially, many of our experiments were based on spiked serum samples, but later it was known that HMGB1 could potentially be falsely elevated in serum samples after the clotting prosess. This is decribed in reference number 17:
- Weng, L.; Guo, L.; Vachani, A.; Mesaros, C.; Blair, I.A. Quantification of Serum High Mobility Group Box 1 by Liquid Chromatography/High-Resolution Mass Spectrometry: Implications for Its Role in Immunity, Inflammation, and Cancer. Anal Chem 2018, 90, 7552-7560, doi:10.1021/acs.analchem.8b01175.
Thus, our focus were on plasma samples, and we have no data on recovery in blood serum compared to blood plasma.
4) Have the authors tried other precipitation methods prior to LC/MS-MS? For example:
Ref: Kay, R., Barton, C., Ratcliffe, L., Matharoo-Ball, B., Brown, P., Roberts, J., Teale, P., Creaser, C. (2008) Enrichment of low molecular weight serum proteins using acetonitrile precipitation for mass spectrometry based proteomic analysis, Rapid Commun. Mass Spectrom., 22, 3255–3260.
Link: https://pubmed.ncbi.nlm.nih.gov/18803344/
This method used with pure recombinant HMGB1 will rapidly inform us on the recovery of the target protein in supernatant, and precipitate if any.
Response:
Thanks for the relevant information and the reference. Our group has successfully used the acetonitrile precipitation for cerebrospinal fluid samples. Unfortunately, our experiments with acetonitrile precipitation of plasma samples were not successful for detection of HMGB1.
We have chosen to add a sentence at the beginning of chapter 2.2.3 Precipitation of HAPs (B), line 211-215:
“Because of HMGB1 being a rather small protein, our initial idea for HAP removal was with acetonitrile (ACN) precipitation as reported for human serum [33] and cerebrospinal fluid [34]. We soon abandoned ACN precipitation because it resulted in non-detectable levels of HMGB1, likely due to co-precipitation with other proteins or poor solubility at the higher concentration levels of ACN that were required for HAP removal.”
Two references were added to the manuscript:
- Kay, R.; Barton, C.; Ratcliffe, L.; Matharoo-Ball, B.; Brown, P.; Roberts, J.; Teale, P.; Creaser, C. Enrichment of low molecular weight serum proteins using acetonitrile precipitation for mass spectrometry based proteomic analysis. Rapid Commun Mass Spectrom 2008, 22, 3255-3260, doi:10.1002/rcm.3729.
- Larssen, E.; Brede, C.; Hjelle, A.B.; Oysæd, K.B.; Tjensvoll, A.B.; Omdal, R.; Ruoff, P. A rapid method for preparation of the cerebrospinal fluid proteome. Proteomics 2015, 15, 10-15, doi:10.1002/pmic.201400096.
5) Line 51: ‘Based on our longstanding study…’ have the authors any reference on that?
Response:
Thanks for pointing out the lack of references here. We have now included some more references in the manuscript, and the sentence on line 51 now reads:
“Based on our longstanding study of chronic fatigue [8-11], we postulated that HMGB1 could be an important signaling molecule in the context of sickness behavior.”
The new references are:
- Bårdsen, K.; Nilsen, M.M.; Kvaloy, J.T.; Norheim, K.B.; Jonsson, G.; Omdal, R. Heat shock proteins and chronic fatigue in primary Sjogren's syndrome. Innate Immun 2016, 22, 162-167, doi:10.1177/1753425916633236.
- Brække Norheim, K.; Imgenberg-Kreuz, J.; Jonsdottir, K.; Janssen, E.A.; Syvänen, A.C.; Sandling, J.K.; Nordmark, G.; Omdal, R. Epigenome-wide DNA methylation patterns associated with fatigue in primary Sjögren's syndrome. Rheumatology (Oxford) 2016, 55, 1074-1082, doi:10.1093/rheumatology/kew008.
- Bårdsen, K.; Brede, C.; Kvivik, I.; Kvaløy, J.T.; Jonsdottir, K.; Tjensvoll, A.B.; Ruoff, P.; Omdal, R. Interleukin-1-related activity and hypocretin-1 in cerebrospinal fluid contribute to fatigue in primary Sjogren's syndrome. J Neuroinflammation 2019, 16, 102, doi:10.1186/s12974-019-1502-8.
- Grimstad, T.; Kvivik, I.; Kvaløy, J.T.; Aabakken, L.; Omdal, R. Heat-shock protein 90α in plasma reflects severity of fatigue in patients with Crohn’s disease. Innate Immunity 2020, 26, 146-151, doi:10.1177/1753425919879988.